# Affinities of Terminal Inverted Repeats to DNA Binding Domain of Transposase Affect the Transposition Activity of Bamboo *Ppmar2* Mariner-Like Element

**DOI:** 10.3390/ijms20153692

**Published:** 2019-07-28

**Authors:** Muthusamy Ramakrishnan, Mingbing Zhou, Chunfang Pan, Heikki Hänninen, Kim Yrjälä, Kunnummal Kurungara Vinod, Dingqin Tang

**Affiliations:** 1State Key Laboratory of Subtropical Silviculture, Zhejiang A&F University, Lin’an, Hangzhou 311300, Zhejiang, China; 2Zhejiang Provincial Collaborative Innovation Center for Bamboo Resources and High-Efficiency Utilization, Zhejiang A&F University, Lin’an, Hangzhou 311300, Zhejiang, China; 3Department of Forest Sciences, University of Helsinki, 00014 Helsinki, Finland; 4Division of Genetics, ICAR-Indian Agricultural Research Institute, Rice Breeding and Genetics Research Centre, Aduthurai, Tamil Nadu 612101, India

**Keywords:** Transposon, Mariner-like elements, Terminal inverted repeat, DNA binding domain, Transposase, Transposition activity, *Phyllostachys edulis*, Moso bamboo

## Abstract

Mariner-like elements (MLE) are a super-family of DNA transposons widespread in animal and plant genomes. Based on their transposition characteristics, such as random insertions and high-frequency heterogeneous transpositions, several MLEs have been developed to be used as tools in gene tagging and gene therapy. Two active MLEs, *Ppmar1* and *Ppmar2*, have previously been identified in moso bamboo (*Phyllostachys edulis*). Both of these have a preferential insertion affinity to AT-rich region and their insertion sites are close to random in the host genome. In *Ppmar2* element, we studied the affinities of terminal inverted repeats (TIRs) to DNA binding domain (DBD) and their influence on the transposition activity. We could identify two putative boxes in the TIRs which play a significant role in defining the TIR’s affinities to the DBD. Seven mutated TIRs were constructed, differing in affinities based on similarities with those of other plant MLEs. Gel mobility shift assays showed that the TIR mutants with mutation sites G669A-C671A had significantly higher affinities than the mutants with mutation sites C657T-A660T. The high-affinity TIRs indicated that their transposition frequency was 1.5–2.0 times higher than that of the wild type TIRs in yeast transposition assays. The MLE mutants with low-affinity TIRs had relatively lower transposition frequency from that of wild types. We conclude that TIR affinity to DBD significantly affects the transposition activity of *Ppmar2*. The mutant MLEs highly active TIRs constructed in this study can be used as a tool for bamboo genetic studies.

## 1. Introduction

Transposable elements (TEs) are genomic factors which can move around in the genome [1,2]. They have been employed as tools for genetic investigations in plants to improve their fitness and growth [3,4]. Based on the mechanism of transposition, there are two classes of TEs, Class I (RNA transposons) and Class II (DNA transposons) [5]. Among the Class II transposons, mariner-like elements (MLE) are a superfamily TEs widespread in diverse taxa, including higher animals, plants, fungi, insects, nematodes and fishes [6,7]. The structure of an MLE is simple, comprising of two target site duplicates (TSDs) of dinucleotide origin of thymine and adenine (TA rich), one open reading frame (ORF) encoding a transposase and two terminal inverted repeats (TIR) flanking the ORF region [8]. The TIRs are usually 10–40 bp long and composed of protein binding elements (PBE) which are specific for individual MLE. For example, in the synthetic transposon *Sleeping Beauty* (*SB*), PBE is located in the TIRs of 15 bp long direct repeats (DR) [9]. Also, the PBE of *Frog Prince* (*FP*) is located in the TIRs of 21 bp long DRs [10]. *FP* and *SB* are Tc1/MLEs superfamily members sharing ~50% sequence similarity and used for gene therapy in fish, amphibian and mammalian cell lines [10,11].

The transposition activities of MLE are regulated by the transposase [12], TIRs, and the flanking sequences [13,14,15]. Specific interaction of transposases with the TIRs and the affinity of TIRs to the transposase influence the activity of MLEs and their transposition efficiency [16,17,18,19]. Usually, the first step in transposition occurs with the binding of transposase to the PBE of TIRs. The binding occurs at the DNA binding domain (DBD) of the transposase. Since TIR sequences can vary at each end, the transposase binding has different affinities to these sequences [20]. For instance, the PBE of the rice MLE, *Osmar5* is 17 bp long and composed of two boxes, Box I and Box II. Mutations of both the boxes have affected the affinity of TIRs to the transposase DBD [17]. Furthermore, the transposition frequency of the DNA-transposase complex is altered by the A and T content of TIRs. In the *Drosophila* MLE, *mos1*, the differential AT content of the right and left TIRs (64.3% and 50.0%, respectively), changes the affinity of the right TIR to be 5–10 times higher than that of the left. When the right TIR was replaced by the left, the affinity was found to decrease by more than 50 times. In contrast, when the left TIR was replaced by the right, the affinity was found to shift in the opposite direction by showing an increase by more than 26 times [21,22,23]. Likewise, when the guanines of 3′ TIRs of the ant (*Messor bouvieri*) transposon, *Mboumar9* were replaced by adenines, the TIR affinity was found four times higher [20]. Additionally, the subterminal region, 30–35 bp flanking sequences of the TIR (Sub-TIR) can also affect the transposition efficiency. Longer the sub-TIRs, such as in the case of *Mos1*, transposition activity was found lowered [24,25]. Importance of sub-TIRs was further demonstrated by Yang et al. [18] through site-directed mutagenesis of the 3′ sub-TIRs of the rice MLE, *Osm14NAS*. They found that a mutation of some motifs could repress the transposition activity.

Based on the transposition characteristics, such as random insertions and high frequency of heterogeneous transpositions, several MLEs such as *Hemar* from the flatworm *Himasthla elongata* [26], *Himar1* from the horn fly *Haematobia irritans* [27], *Hsmar1* from humans [28], and *Mos1* [29] have been used as tools for gene tagging and gene therapy. Lampe et al. [30] found that enhanced transposase activity of the mutated *Himar1* was due to the increased affinity of DNA in general. Enhanced DNA-binding activity of hyperactive *SB* element was due to improved cleavage kinetics and increased element mobilisation from host cell chromosomes [31], which dramatically enhanced gene transfer capabilities in vivo in mice [32]. In order to improve *Mos1* transposition frequency, hyperactive *Mos1* transposase versions were generated by site-directed mutagenesis [13,14]. Although several MLEs have been found functional in yeast (*Saccharomyces cerevisiae*), their activities in general in plants are relatively less studied [18,33].

Among the commercially important plants, bamboos adorn a significant place for being a prime source of industrial raw material, contributing substantially to the South Asian economy [34,35,36]. Worldwide, there are more than 1642 cultivated bamboo species belonging to 75 genera adapted to diverse climatic regions, especially confined to China, India, Japan, Korea, Myanmar and Australia (https://www.inbar.int/, last access on: 24 July 2019). Among these, moso bamboo (*Phyllostachys edulis*) is an important commercial species in China that generates an equivalent of about 5 billion US dollars from industries including textile, timber and food [37]. Characterized by its rapid growth, unique biomass turnover and adaptation to temperate conditions, moso bamboo not only provides raw materials for paper and rayon industries but is also popular for its unique timber strength.

Reproductive behaviour in bamboos has invoked scientific curiosity for a long time because flowering intervals range from several years to more than a hundred years between the species, and plants often die after fruiting [38]. The moso bamboo has a long lifespan and shows sporadic flowering rather than the gregarious flowering as seen in other bamboo species. Understanding the genetic and molecular mechanism of flower development and rapid growth [39] remains the main challenge, with conclusive answers still elusive [37,40]. Similar to any cultivated species, moso bamboo suffers biotic and abiotic stresses under cultivation. Being predominantly clonally propagated with an unpredictable and extended breeding cycle, relatively little has taken place in terms of genetic improvements for yield, quality and stress tolerance. Presently, the whole genome information of moso bamboo is available, and a chromosome level genome database [41] and BambooNET database (http://bioinformatics.cau.edu.cn/bamboo, last access on: 24 July 2019) [42] have been developed. These novel tools can drive investigations in unravelling the reproductive and vegetative biology of moso bamboo. Although the genetic transformation system in moso bamboo remains under establishment, transposon-based mutagenesis can be employed as an additional tool to manipulate the moso bamboo genome [15] in the investigations of genetic routes of bamboo biology.

Among the reported transposons in bamboo, 82 MLEs from 44 bamboo species belonging to 38 genera have been characterized [43]. A few MLEs found in the moso bamboo genome [43,44] indicated highly conserved transposases [45]. Two active full-length MLEs, *Ppmar1* and *Ppmar2*, isolated from moso bamboo that contained perfect TIRs and intact transposases [46], showed size differences, with *Ppmar*1 being longer and belonging to A2 subfamily, and the shorter *Ppmar2* belonging to C subfamily [46]. Both these elements have shown transposition activity in *Arabidopsis thaliana* and established their preferential insertion to the TA rich regions with random insertions in the host genome [44]. Additionally, fourteen different hyperactive *Ppmar1* transposase variants generated by single amino acid substitutions were shown to increase the transposition activity of *Ppmar1* MLEs [15]. When a non-autonomous transposon *Ppmar1NA* system was generated by excising the transposase gene, *Ppmar1* transposase variants were shown to promote element excision and reintegration of the non-autonomous *Ppmar1NA* in the yeast genome at TA dinucleotide sequences. The most hyperactive transposase variant S171A was found to induce 10-fold more active excisions in yeast [15].

Notwithstanding the effects of transposases, MLE transposition efficiency is also dependent on the binding affinities of TIRs to DBD [18,19,20], as well as to the length of sub-TIRs. However, information on the TIR variants and their influence on transposition in the moso bamboo MLEs are not yet available. This study aims to fill this gap by mutating the TIRs of *Ppmar2* through site-directed mutagenesis to alter transposition activity and to quantify their affinities to DBD of *Ppmar1* and *Ppmar2* transposases. The affinity variants were investigated by electrophoretic mobility shift assay (EMSA), also known as “gel mobility shift assay”, while the transposition activity of shortened *Ppmar2* mobilized by the mutant TIRs and catalysed by *Ppmar1* and *Ppmar2* transposases was investigated by yeast transposition assays.

## 2. Results

### 2.1. The Identification of Box I and II in Ppmar2-TIRs and Development of Mutant Sequences

The non-autonomous *Ppmar2NA*-TIR1, lacking the transposase gene, was developed from the wild *Ppmar2* sequences (Figure 1) and two conserved domains, Box I and Box II, were identified within the TIRs by comparative analysis with TIRs of six other MLE sequences (*Osmar5*, *Ammar1*, *Famar1*, *Hsmar1*, *Mos1* and *Himar1*). The conserved domain sequence of Box I was TCCCA and of Box II was GGCG (Figure 2). It was found that in Box1, cytosine (C) was more conserved than T and A, whereas in Box II, guanine (G) was more conserved than C. Seven base mutations designed within the two boxes of *Ppmar2NA*-TIR1, four in Box I and three in Box II (Table 1), had 19 nucleotide substitutions in total. Ten substitutions were made to A, seven to T, one each to G and C. In Box I, one T was substituted by A, five C were substituted by T, two C by A and three A were substituted by one C, T and G. In Box II, four Gs were substituted by A and three C by three As.

### 2.2. Affinity Analysis of Ppmar2NA-TIRs to Ppmar1 and Ppmar2 Transposases

The EMSA analysis revealed that synthesized *Ppmar2NA*-TIRs and DBDs based on *Ppmar1* and *Ppmar2* sequences showed differential affinity, with the former showing less affinity and the latter having a high affinity (Figure 3). In general, the binding of *Ppmar1NA*-TIRs occurred at higher concentrations of *Ppmar1*-DBD than for *Ppmar2*-DBD. Therefore, the concentration of *Ppmar1*-DBD was varied as 0, 200, 300, 400 and 500 µM, while that of *Ppmar2*-DBD was set as 0, 100, 120, 140, and 150 µM. In both cases, *Ppmar2NA*-TIRs was kept at 1 µM.

*Ppmar2NA*-TIRs exhibited a very weak affinity to *Ppmar1*-DBD even at higher concentrations. In most of the TIRs, a non-specific pattern across concentrations was observed. A relatively strong affinity was observed in two mutants, *Ppmar2NA*-TIR6 and *Ppmar2NA*-TIR7, also at 500 µM concentration. Unlike with *Ppmar1*-DBD, we found that almost all *Ppmar2NA*-TIRs were bound to *Ppmar2*-DBD when the concentration of *Ppmar2*-DBD was above 150 µM. There was a regular pattern of affinity in this case, which increased with the concentration. This indicated that 0–200 µM concentrations of *Ppmar2*-DBD were suitable for detection of the affinity between *Ppmar2NA*-TIRs and *Ppmar2*-DBD.

The DBD affinities of mutant *Ppmar2NA*-TIRs varied significantly from the wild type (*Ppmar2NA*-TIR1) and indicated an increased binding in all the cases. The binding was highest in *Ppmar2NA*-TIR6 followed by *Ppmar2NA*-TIR7 at 150 µM concentrations. Among the mutants, *Ppmar2NA*-TIR3 was observed to have relatively less affinity shift towards elevated concentrations. In the remaining mutants, such as *Ppmar2NA*-TIR2, *Ppmar2NA*-TIR4 and *Ppmar2NA*-TIR8 the affinity gradient was almost similar to that of the wild type. In general, at the 150 µM concentration level of *Ppmar2*-DBD, the *Ppmar2NA*-TIRs showed the highest level of affinities in both wild types and mutants except for *Ppmar2NA*-TIR8. The highest affinity in *Ppmar2NA*-TIR8 was detected at 140 µM concentration.

Analysis of affinity shift in the mutants from the wild type, with respect to their individual mutations, indicated that Box I mutations were relatively more non-specific than Box II mutations (Table 2). The non-specificity was particularly apparent with *Ppmar1*-DBD compared to *Ppmar2*-DBD. The Box I mutations were predominantly C→T substitutions and C→A. However, Box II mutations were significantly different from those of Box I because at least two mutants, *Ppmar2NA*-TIR6 and *Ppmar2NA*-TIR7, exhibited affinity consistency across transposases. In Box II, there were four G→A substitutions, three C→A substitutions and one C→T substitution. Across Box I and Box II, it is evident that C→A substitutions have consistently resulted in higher affinity with *Ppmar2* transposase. Furthermore, the G substitutions in Box II either to A or T seemed to improve the TIRs affinity for transposases from both the moso bamboo MLEs. However, when there were too many G substitutions in Box II, the affinity shift was found reversed for *Ppmar1*-DBD while it was reduced for *Ppmar2*-DBD, as found in the mutant *Ppmar2NA*-TIR8.

### 2.3. The Influence of Affinity Contrasting Mutant TIRs on the Transposition Frequency

Considering that the sub-terminal sequences of TIRs are known to affect the transposition of MLEs in the yeast, sub-TIRs of *Ppmar2NA* were deleted to avoid ambiguity in the yeast transposition assay. Three TIRs were used for the transposition assay: Two mutants (*Ppmar2NA*-TIR3 (C657T-A660T) with a weak affinity; and *Ppmar2NA*-TIR6 (G669A-C671A) with a strong affinity); and the wild-type *Ppmar2NA*-TIR1. The vectors pWL89a-*Ppmar2NA*-TIR1, pWL89a-*Ppmar2NA*-TIR3 and pWL89a-*Ppmar2NA*-TIR6 were each co-transformed into yeast cells, together with pAG413gal-Tpase1 and pAG413gal-Tpase2. When the transformed colonies with TIR alone and without transposase were grown on a complete supplement mixture (CSM) medium without histidine and uracil, but adenine sufficient (CSM-his-ura) and adenine deficient (CSM-ade-his-ura), there was normal growth of colonies in the former and no colonies developed in the latter. Alternatively, when the double-transformed colonies possessing TIRs and transposase were grown on the same media, there was colony development in both the cases (Figure 4). Adenine is essential for colony growth, and it should be provided either from the medium or be synthesized internally by the yeast cells. In the transformed yeast cells, however, the adenine synthesis is arrested by silencing the key enzyme *ADE2* coding for phosphoribosylaminoimidazole carboxylase, in the purine synthetic pathway. In this case, the silencing was achieved by placing the TIR sequences within the *ADE2* gene of the plasmid vector pWL89a, which was used for transformation. The colony growth in the adenine deficient medium of the double transformants indicated a de novo *ADE2* activity, resulting in systemic adenine synthesis. To become active, the TIR sequences need to be cleaved out of the *ADE2* exons, which is possible only if transposase binds to TIRs initiating transposition activity. The proportion of *ADE2* revertants in the yeast assay indicated the transposition efficiency of the transposases.

In the affinity contrasting mutant TIRs, *Ppmar2NA*-TIR3 (C657T-A660T) and *Ppmar2NA*-TIR6 (G669A-C671A), more frequent transposition was catalyzed by *Ppmar1* transposase than *Ppmar2* transposase, whereas for the wild type TIR *Ppmar2NA*-TIR1, the relative transposition frequency of both the transposases was similar (Figure 5). In the low-affinity TIR mutant *Ppmar2NA*-TIR3, the *Ppmar1* transposase produced an almost similar frequency of transposition as that of the wild type, while the *Ppmar2* transposase resulted in half the frequency of transposition than the wild type. Nevertheless, in the high-affinity mutant TIR, *Ppmar2NA*-TIR6, the transposases of *Ppmar1* and *Ppmar2* resulted in 1.5 and 1.8 times higher transposition activity than the wild type, respectively. In agreement with the in vitro affinity pattern (Figure 3), strong affinity *Ppmar2NA*-TIRs had a high transposition activity, while those with weak affinity indicated a low transposition activity.

## 3. Discussion

Moso bamboo, the most valued bambusoid on earth, has a genome size of 2075 Mb and contains 59% of TEs [47], although very few TEs have been characterized so far. TEs are potential tools for genetic studies because of their ability to move around the genome creating random insertions. This behaviour of the TEs can be exploited to create mutations within genes to silence them, thereby enabling the studies of gene functions [1,2,3,4]. Transposition of the TEs from the silenced gene can reactivate them on a later stage. Since TEs are natural elements, they play a significant role in the evolution of crop species and drive the genomes to adapt themselves to unfavorable environments by creating beneficial mutations [48]. Among the TEs, MLEs are Class II DNA transposons abundantly available in the living organisms. Of the several MLEs available in the bamboo genomes, two MLEs are characterized in moso bamboo, namely, *Ppmar1* and *Ppmar2* [45]. To use them as a potential tool for genetic investigations in moso bamboo, there is a need to document the efficiency of these elements. There are several factors that determine the transposition efficiency of MLEs, such as internal and external factors [7,15]. Internal factors are primarily structural features such as the TIRs, sub-TIRs and the transposase sequences. There can be several external factors such as internal and external cellular environment, quantum and duration of accumulation transposases, age and metabolic state of the cell, and the abundance of active TEs in the cells [49].

TIRs of the MLEs vary significantly between different classes and families [9,10,11]. However, there are certain regions that remain highly conserved within TIRs, especially at the PBE site that plays an important role in transposition activity. Fundamentally, transposition requires a transposase to bind to the PBE of the TIRs triggering the activity [9,10,11]. Binding requires a mutual affinity between the PBE and the transposase DBD. Therefore, altering the nucleotide balance in the TIR conserved domain boxes could bring about variations in affinity. We were able to construct mutations in the conserved boxes I and II of the *Ppmar2* element by random nucleotide substitutions. The effect of these mutations was tested by stripping the transposase off the TIRs and allowing them to bind to the DBD domain in vitro to assess the affinity. In the in vitro system, the binding affinities of transposons to the DBD of both *Ppmar1* and *Ppmar2* transposases varied with different mutation levels in the non-autonomous elements of *Ppmar2NA*. *Ppmar1* transposase was less specific to mutations in Box I than that in Box II, while *Ppmar2* transposase showed high affinity with both boxes. This is in agreement with the previous report [17,18], where multiple variants of the *Osmar5* TIRs showed significant effects on the binding of *Osmar5N* with respect to the mutations they carried. The significant impact on the binding of *Ppmar1* and *Ppmar2* transposases to multiple variants of both Box I and Box II observed in our study indicates that both boxes are critical in prescribing the affinity. Since *Ppmar2*-DBD is the native transposase domain of the *Ppmar2* element, it is reasonable to assume that this domain will have a preferably greater affinity than the *Ppmar1*-DBD.

Previously, we reported the development of *Ppmar1* transposase variants by single amino acid substitutions based on homology analysis with the other functional MLEs [15]. We generated non-autonomous transposons, *Ppmar1NA* and *Ppmar2NA,* by deleting the transposase gene from *Ppmar1* and *Ppmar2* sequences, while retaining the 5′- and 3′-TIRs along with their corresponding sub-terminal sequences [15]. The *Ppmar1* transposase variants catalysed *Ppmar1NA* to transpose in a yeast system, but *Ppmar2NA* showed false-positive insertions, even when transposase was absent. Transposition occurred in TA rich regions, and transposase variant S171A was more active than wild type transposase. Apart from the transposase, the TIRs and their flanking sequences, chromatin status, DNA methylation, and host proteins might affect the transposition activity [18,50]. In the current study, TIR variants were developed to increase the transposition frequency of transposon, based on the alignment of *Ppmar2* TIRs with other TIRs such as *Osmar5*, *Ammar1*, *Famar1*, *Himar1*, *Hsmar1* and *Mos1*.

Moreover, we deleted the sub-TIRs of *Ppmar2NA* to overcome the difficulty in the yeast transposition assay. With respect to the nucleotide substitutions, the substitution of G with A and T in Box II was found to improve the affinity tremendously, suppressing the transposase specificity between *Ppmar1* and *Ppmar2*. Further, the C→A substitutions indicated a positive effect on binding affinity across the boxes and transposases. Three base-pair substitutions had reduced or no significant impact on the affinity, whereas two base-pair substitutions increased the interaction of DBD and *Ppmar2NA*-TIRs. This is in agreement with a previous report on rice [17], where two or more base-pair substitutions were found to decrease the level of interaction between TIRs and transposase. We have identified two high-affinity TIRs, *Ppmar2NA*-TIR6 and *Ppmar2NA*-TIR7, which showed almost equal affinity towards *Ppmar1* and *Ppmar2* transposases. Our data support a model of the cross-mobilisation of different MLEs [17,18]. In rice MLEs, one specific *Osmar* transposase can interact with the TIRs of other *Osmar* elements in the same clade. For example, *Osmar5N* was able to bind to the TIR sequences of rice MLEs from the same clade [17]. In our study, *Ppmar2NA*-TIRs could also be bound to the DBD of *Ppmar1* transposase, which indicated that the *Ppmar2NA* might be mobilized by *Ppmar1* transposase catalysis as well. It would be ideal to test their improved general affinity with other transposase systems. These elements can be further employed in genetic studies, not only in moso bamboo but also in other organisms.

In the yeast transposition system, the *Ppmar2NA*-TIR6 transposon showed 1.8 times higher transposition activity than the wild type *Ppmar2NA*-TIR1 with *Ppmar1* transposase, which was 1.5 times higher with *Ppmar2* transposase. Although there was a clear advantage of increased binding affinity for *Ppmar2* transposase to the TIR variants indulged in this study, the efficiency of transposition was relatively higher with *Ppmar1* transposase. The increased efficiency with *Ppmar1* transposase could be attributed to its natural relative efficiency over the *Ppmar2* enzyme. There are various factors that can affect the efficiency of the transposon, which may include affinity level, binding non-specificity and the size of the transposon itself [49]. Whether higher efficiency of *Ppmar1* can be attributed to its size—*Ppmar1* is a longer element than *Ppmar2*—is a question for further investigation. Among the multiple variants of the *Ppmar2NA*-TIRs, affinity towards *Ppmar2*-DBD was stronger for *Ppmar2NA-TIR6* (G669A-C671A), while the same was weaker for *Ppmar2NA-TIR3* (C657T-A660T). High-affinity TIRs could produce a high frequency of transposition, as observed with the *Ppmar2NA*-TIR6 element. Perhaps *Ppmar2NA*-TIR6 could have had a stronger interaction with DBD than the wild type, facilitating catalysis of more frequent transposition either by *Ppmar1* transposase or *Ppmar2* transposase. On the other hand, the weak affinity *Ppmar2NA*-*TIR3* had a reduced activity than the wild type *Ppmar2NA*-TIR1 and resulted in poor catalysis by both the enzymes. The broader biological implications of *Ppmar2NA-TIRs* needs to be addressed in future investigations. Furthermore, to validate the hyperactivity of mutated *Ppmar2NA-TIRs*, they need to be examined in model plants such as foxtail millet, rice or *Arabidopsis*. This could help to develop actively modified transposons as tools for genetic manipulations and bamboo improvement.

## 4. Materials and Methods

### 4.1. The Synthesis of Ppmar2NA-TIRs

The *Ppmar2* used was a full-length MLE, with perfect 27-bp TIRs, the 5’-TIR (CTC CCT CCG TCC CAG TAT AAC TTT TTT) and the 3’-TIR (AAA AAA GTT ATA CTG GGA CGG AGG GAG) [46]. The end-to-end TIRs were synthesized without the intervening transposase sequences by Sangon Biotech (Shanghai, China) and named as non-autonomous *Ppmar2NA*-TIR1. The synthesized sequence was 58 bp long with a structure of TA CTC CCT CCG TCC CAG TAT AAC TTT TTT AAA AAA GTT ATA CTG GGA CGG AGG GAG AT, including two target site duplications (TSD). The structure of *Ppmar2* (autonomous) and *Ppmar2NA*-TIR1 (non-autonomous) is shown in Figure 1.

### 4.2. Identification of Conserved Domains and Mutations in Ppmar2NA-TIRs

The non-autonomous *Ppmar2NA*-TIR1 was aligned with TIRs sequences of *Osmar5* (Accession No. GQ382183) from rice [33], *Ammar1* (Accession No. AY155490) from European honey bee (*Apis mellifera*) [51], *Famar1* (Accession No. AY155492) from the earwig (*Forficula auricularia*) [52], *Himar1* (Accession No. U11646) from horn fly (*Haematobia irritans*) [50,53], *Hsmar1* (Accession No. U52077) from human genes [54], and *Mos1* (Accession No. X78906.1) from *Drosophila mauritiana* [55]. Two conserved domains were found in *Ppmar2NA*-TIR1 and named as Box I and Box II according to literature (Figure 2). Using the conserved domain sequences of *Ppmar2NA*-TIR1 and based on our previous reports of *Ppmar1* and *Ppmar2* transposition activity in *Arabidopsis thaliana* [44], seven base mutations were designed in the two Boxes of *Ppmar2NA*-TIR1 (Table 1). The mutated TIRs were synthesised by Sangon Biotech (Shanghai, China) and named *Ppmar2NA*-TIR2 to *Ppmar2NA*-TIR8, respectively.

### 4.3. Construction of Ppmar2NA Donor Vector

The *Ppmar2NA* was inserted into the *XhoI* site of pWL89a vector (kindly provided by Dr Susan R Wessler from University of Georgia, USA) resulting in pWL89a-*Ppmar2NA*. The inserted product was transformed into *E. coli* DH5α and positive clones were screened.

### 4.4. Mutagenesis of Ppmar2NA-TIRs

The mutagenesis of *Ppmar2NA*-TIRs was performed with the QuikChange lightning site-directed mutagenesis kit (Agilent Technologies, Santa Clara, CA, USA) using four pairs of primers. The name of the primers and their sequences are given in Table 3. Mutagenesis reactions in 25 µL mix contained 100 ng of the template of *Ppmar2NA* donor vector, 2 µM of each primer, and 0.5 µL of QuikChange lightning buffer. The plasmid vector pWL89a-*Ppmar2NA* was used as the template for site-directed mutagenesis. To confirm the presence of the targeted mutation, all plasmids were sequenced with specific primers.

### 4.5. The Extraction and Synthesis of Transposases Ppmar1-DBD and Ppmar2-DBD

The DBDs of *Ppmar1* and *Ppmar2* transposases used in the current study contained 22 amino acids each, MTI EDV SSR LGI SKS RIQ RYL K for the former and TTI RDL AGA LNI SKS TLF RQM K for the latter. The homologous alignment of the DBD amino acid sequences of transposase (Figure 6) indicated a helix-turn-helix (HTH) motif [45]. The DBD sequences were synthesised by Sangon Biotech (Shanghai, China) and named as *Ppmar1*-DBD and *Ppmar2*-DBD, respectively.

### 4.6. Construction of Transposase Expression Vectors

In order to construct the transposase expression vectors, full-length transposases of *Ppmar1* and *Ppmar2* were cloned between *Not1* and *EcoR*V sites of the pMD-18-T vector (Takara, Japan). The *Ppmar1* and *Ppmar2* transposase ORFs were amplified respectively by primer pairs, Tpase1-F + Tpase1-R andTpase2-F + Tpase2-R, both containing *Not*1 and *EcoR*V restriction sites (Table 4). The PCR product was digested by *Not*1 and *EcoR*V. The pAG413gal-ccdB vector (kindly provided by Dr Susan R Wessler from University of Georgia, USA) was digested by *Not*1 and *EcoR*V and the digested PCR products were ligated to the large fragment of the pAG413gal-ccdB vector. The transposases of *Ppmar1* and *Ppmar2* sequences replaced the *ccdB* region in the pAG413gal-ccdB vector and were named as pAG413gal-Tpase1 and pAG413gal-Tpase2, respectively.

### 4.7. Affinity Analysis of Ppmar2NA-TIRs to Ppmar1-DBD and Ppmar2-DBD

The single-strand *Ppmar2NA*-TIRs including their mutants (TIR1-TIR8) were dissolved in ultra-pure water to an initial concentration of 10 µM. The mix was then incubated at 100°C in a boiling water bath for 10 min for denaturation and renatured into double TIRs by natural cooling. The *Ppmar1*-DBD and *Ppmar2*-DBD were dissolved separately in protein buffer (100 mM NaCl, 10 mM Tris, 10% glycerol, pH 8.0) to a concentration of 1 mM. The concentration of different mutated *Ppmar2NA-TIRs* was set at 1 µM, while the concentrations of *Ppmar1*-DBD and *Ppmar2*-DBD were set at 0, 100, 120, 140, 150, 200, 300, 400, or 500 µM. The double TIRs (1 µM) and the diluted *Ppmar1*-DBD and *Ppmar2*-DBD with their corresponding concentrations were fully mixed (10 µL) and incubated at 65 °C for 20 min and then cooled down. The mixture was separated using 18% non-denaturing polyacrylamide gel electrophoresis (PAGE) (BioRad, Hercules, CA, USA), followed by silver staining [56]. The gel images were quantified for the band intensity using ImageJ v.1.52o [57].

### 4.8. Estimation of the Transposition Frequency

The transposition assay was performed using the yeast haploid strain DG2523 (*MATalpha ura3-167 trp1-hisG leu2-hisG his3-del200 ade2-hisG*) (kindly provided by David Garfinkel, University of Georgia, USA). Two plasmids, pWL89a-*Ppmar2NA* and pAG413gal-Tpase1 (or) pAG413gal-Tpase2 were co-transformed into DG2523 and the transformed colonies were selected by growing them for ten days at 30 °C in a complete supplement mix (CSM) devoid of histidine and uracil (CSM-his-ura) in presence of 2% galactose. Each galactose-induced yeast cell colony was suspended in 50 μl of water and plated on CSM-ade-his-ura medium which additionally lacks adenine. After a seven-day incubation at 30 °C, the ADE2 revertant colonies were counted for all mutant constructs [15]. The excision frequencies were computed as the number of revertant per live yeast cell, based on the density of live cells in the suspension used for plating. The transposition assay workflow is shown in Figure 7.

## 5. Conclusions

To conclude, we have found that the *Ppmar2NA*-TIRs developed in this study can be mobilized in the yeast cell. Two conserved domain boxes were identified in the TIRs with varying affinities to the DBDs of *Ppmar1* and *Ppmar2* transposases. The affinity of TIRs to DBD could significantly influence the transposition activity of MLEs in certain elements with non-specific affinity to transposases. Hyperactive *Ppmar2NA*-TIR mutants could be developed to use a genetic tool for the construction of bamboo mutant libraries for gene identification and subsequently for bamboo breeding. Further, they can also be used in mutagenesis studies in other organisms as well.

## Figures and Tables

**Figure 1 ijms-20-03692-f001:**
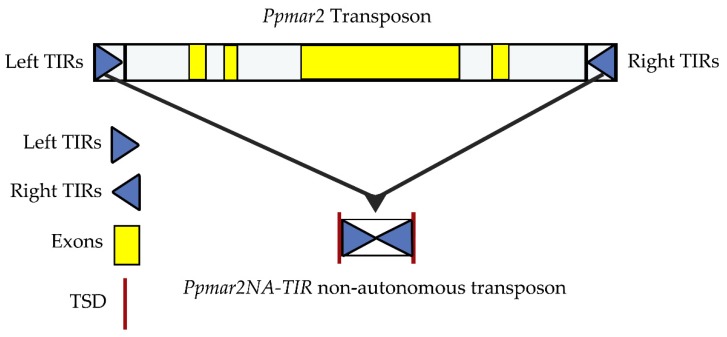
Structure of the *Ppmar2* transposon and the non-autonomous transposon *Ppmar2NA*-TIRs (TIRs—terminal inverted repeats). Yellow bars represent the exons of *Ppmar2* transposase, blue triangles the TIRs of *Ppmar2* transposon, and dark red vertical lines the target site duplications (TSD).

**Figure 2 ijms-20-03692-f002:**
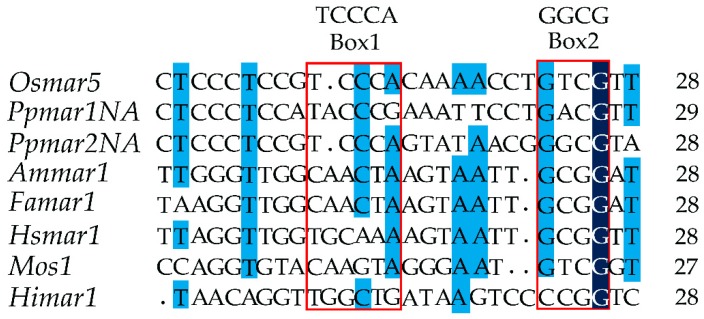
The alignment of TIRs of Mariner-like elements (MLEs). *Osmar5*, AP008207 of *Oryza sativa*; *Ammar1*, U19902 of *Apis mellifera*; *Famar1*, AY226507 of *Forficula auricularia*; *Hsmar1*, U52077 of *Homo sapiens*; *Mos1*, X78906 of *Drosophila mauritiana*; *Himar1*, U11646 of *Haematobia irritans*.

**Figure 3 ijms-20-03692-f003:**
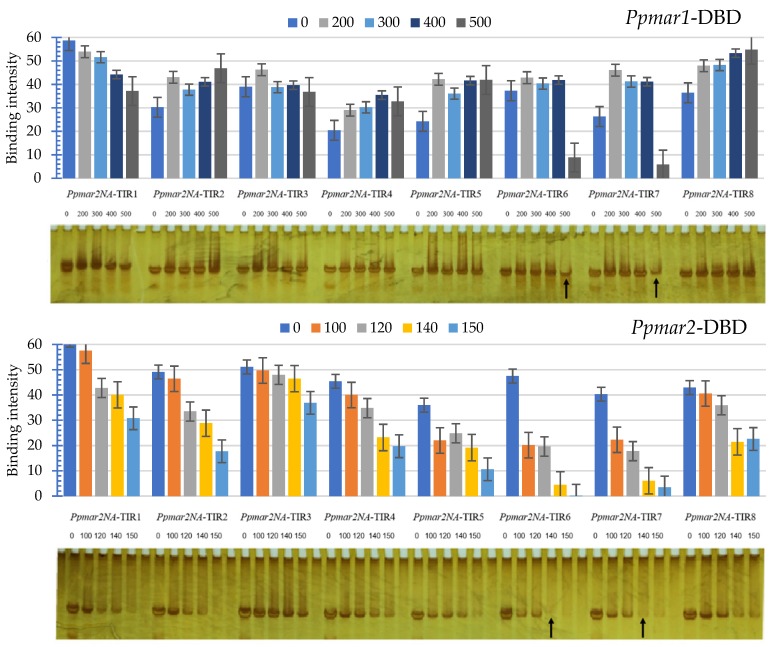
Electrophoretic mobility shift assays (EMSA) of mutated *Ppmar2NA*-TIRs 1–8, obtained with different concentrations of *Ppmar1*-DBD and *Ppmar2*-DBD. The concentration varied between 0 and 500 µM for the former and 0 and 150 µM for the latter. Black arrows indicate the highest interactions of mutated *Ppmar2*-TIRs and DBD forming the bound complex, as shown by the relative scantiness of *Ppmar2NA*-TIRs in the corresponding lanes. The error bars indicate standard errors.

**Figure 4 ijms-20-03692-f004:**
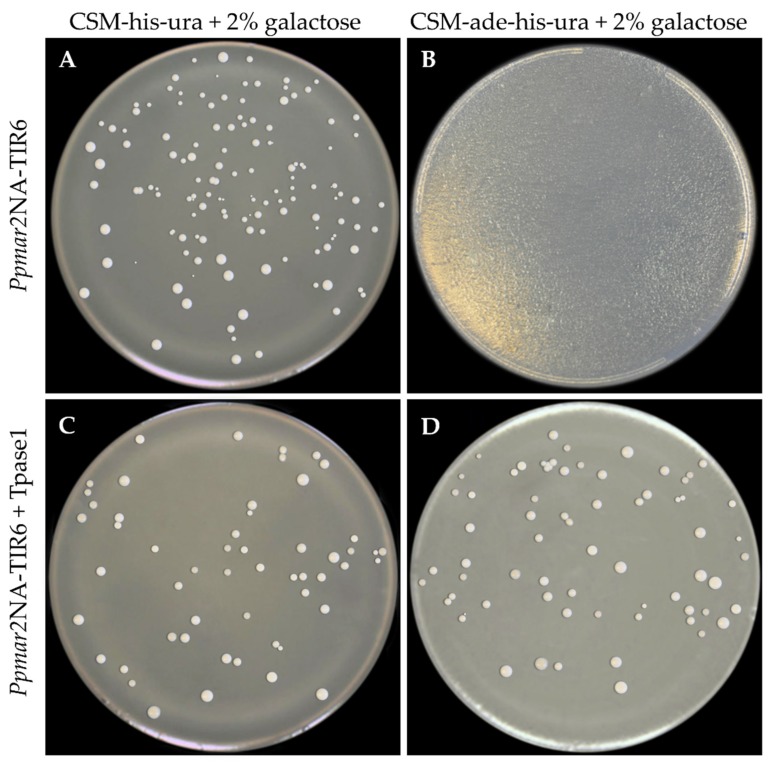
Yeast transformation assay for the active mutant TIR *Ppmar2NA*-TIR6, showing active colony growth in the presence of adenine, irrespective of the presence of *Ppmar1* transposase (Tpase1) ((**A**), transposase absent; (**C**), transposase present). When grown on adenine free medium, no colonies developed when transposase was absent (**B**), but colonies developed when transposase was present (**D**). The transposase could initiate the transposition of the TIR sequences out of the *ADE2* gene, thereby reactivating the *ADE2* resulting in de novo adenine synthesis.

**Figure 5 ijms-20-03692-f005:**
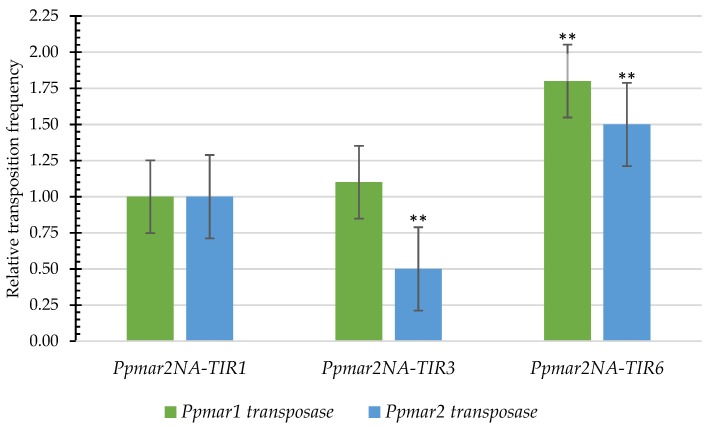
Relative transposition frequencies of mutated *Ppmar2NA*-TIRs triggered by *Ppmar1* and *Ppmar2* transposases. The Y-axis indicates the ratio of the mean excision frequency of six independent experiments with *Ppmar2NA* mutants, to that of the wild type *Ppmar2NA*-TIR1. The symbol ** indicates a statistically significant (*p* < 0.01) difference with the wild type *Ppmar2NA*-TIR1. Error bars indicate standard errors.

**Figure 6 ijms-20-03692-f006:**
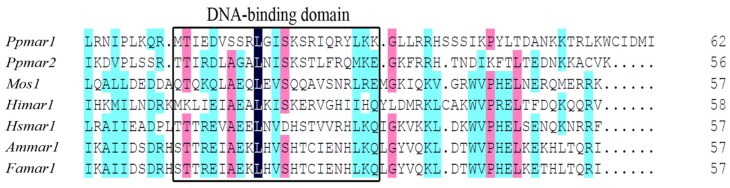
The homologous alignment of amino acid sequences of DNA binding domain (DBD) of *Ppmar1* transposase and the DBD of *Ppmar2* transposase in the following DBDs: *Mos1* from *Drosophila mauritiana*, *Himar1* from horn fly, *Hsmar1* from human genes, *Ammar1* from European honey bee, and *Famar1* from earwig.

**Figure 7 ijms-20-03692-f007:**
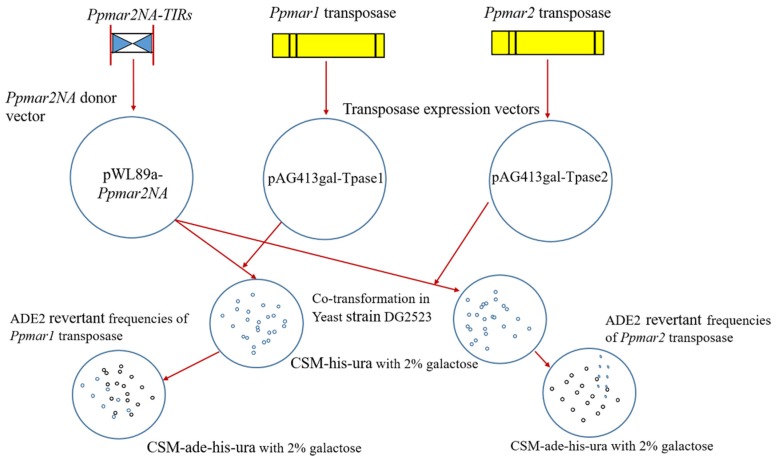
The vector constructs and workflow of *Ppmar2NA*-TIRs transposition assay in yeast cells (*Saccharomyces cerevisiae*). The *Ppmar2NA*-TIRs were inserted to the pWL89a vector, resulting in the vector pWL89a-*Ppmar2NA*. The *Ppmar1* and *Ppmar2* transposases were separately inserted to the pAG413gal-ccdB vector, resulting in pAG413gal-Tpase1 and pAG413gal-Tpase2, respectively. The yeast cells were grown on medium CSM-his-ura with 2% galactose. The co-transformation of pWL89a-*Ppmar2NA* and pAG413gal-Tpase1 vectors and co-transformation of pWL89a-*Ppmar2NA* and pAG413gal-Tpase2 vectors were inserted into yeast strain DG2523. ADE2 revertant cells were grown on medium CSM-ade-his-ura with 2% galactose.

**Table 1 ijms-20-03692-t001:** The *Ppmar2NA*-TIRs and their sequences with mutations details. The mutated bases are marked by the red font. Bold and underlined sequences on the left and right within each TIR represent Box1 and Box2, respectively.

S. No.	Name of TIR	Sequence of TIR Variants
1	*Ppmar2NA*-TIR1	TACTCCCTCCG**TCCCA**GTATAACG**GGCG**TATAAAAAAATTT
2	*Ppmar2NA*-TIR2	TACTCCCTCCG**ATACA**GTATAACG**GGCG**TATAAAAAAATTT
3	*Ppmar2NA*-TIR3	TACTCCCTCCG**TTCCT**GTATAACG**GGCG**TATAAAAAAATTT
4	*Ppmar2NA*-TIR4	TACTCCCTCCG**TTTCG**GTATAACG**GGCG**TATAAAAAAATTT
5	*Ppmar2NA*-TIR5	TACTCCCTCCG**TACTC**GTATAACG**GGCG**TATAAAAAAATTT
6	*Ppmar2NA*-TIR6	TACTCCCTCCG**TCCCA**GTATAACG**AGAG**TATAAAAAAATTT
7	*Ppmar2NA*-TIR7	TACTCCCTCCG**TCCCA**GTATAACG**GG****AT**TATAAAAAAATTT
8	*Ppmar2NA*-TIR8	TACTCCCTCCG**TCCCA**GTATAACG**AAAA**TATAAAAAAATTT

**Table 2 ijms-20-03692-t002:** The details of the *Ppmar2NA*-TIRs interaction with different concentration of *Ppmar2*-DBD and *Ppmar1*-DBD transposase.

*Ppmar2*-TRIs	Nucleotide Substitutions *		Affinity Shift (%)	
	Box I	Box II	Ppmar1-DBD @ 500 µM	Ppmar2-DBD @ 150 µM
*Ppmar2NA*-TIR1	++	++	0.00	0.00
*Ppmar2NA*-TIR2	T→A, C→T, C→A	++	−26.10	42.38
*Ppmar2NA*-TIR3	C→T, A→T	++	1.03	–19.89
*Ppmar2NA*-TIR4	C→T (2), A→G	++	11.81	35.93
*Ppmar2NA*-TIR5	C→A, C→T, A→C	++	−12.75	65.55
*Ppmar2NA*-TIR6	++	G→A, C→A	76.20	99.55
*Ppmar2NA*-TIR7	++	C→A, G→T	84.23	88.87
*Ppmar2NA*-TIR8	++	G→A (3), C→A	−47.47	26.64

++, wild type; * values in brackets show the number of times that particular nucleotide substitution occurred in the mutant; affinity shift is the change in affinity in mutants from the wild type and expressed as the percentage over wild type.

**Table 3 ijms-20-03692-t003:** The primers and their sequences used for site-directed mutagenesis of *Ppmar2NA*-TIRs.

S. No.	Primer	Sequence (5′-3′)
1	XT-C657T-A660T-F	CGAGTACTCCCTCCGTTCCTGTATAACGGGCGTATA
2	XT-C657T-A660T-R	TATACGCCCGTTATACAGGAACGGAGGGAGTACTCG
3	XT-G669A-C671A-F	CCCTCCGTCCCAGTATAACGAGAGTATAAAAAAATTTCAGAGAC
4	XT-G669A-C671A-R	GTCTCTGAAATTTTTTTATACTCTCGTTATACTGGGACGGAGGG
5	XT-T1459A-G1462A-F	TTTTTTATACGCCCGTTATACAGGAACGGAGGGAGTACTCG
6	XT-T1459A-G1462A-R	CGAGTACTCCCTCCGTTCCTGTATAACGGGCGTATAAAAAA
7	XT-G1448T-C1450T-F	GGACAGCAGAAATTTTTTTATACTCTCGTTATACTGGGACGGAGG
8	XT-G1448T-C1450T-R	CCTCCGTCCCAGTATAACGAGAGTATAAAAAAATTTCTGCTGTCC

**Table 4 ijms-20-03692-t004:** The detailed sequences of primers used in the study to amplify *Ppmar1* and *Ppmar2* transposases.

S. No.	Primer Name	Sequence (5′-3′)
1	Tpase1-F	AGAATGCGGCCGCAAAAAAATGGCTGACCCAATAGATTCGCGGCCGC
2	Tpase1-R	GACTGATATCTGCTGCTGCAAAAGAGTAACGAT ATC
3	Tpase2-F	AGAATGCGGCCGCAAAAAAATGGCGAATTTGGACCTAAATCGCGGCCGC
4	Tpase2-R	GACTGATATCCTAATTGATGTACACAATTGGATATC

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
