# Peer review of "Affinities of Terminal Inverted Repeats to DNA Binding Domain of Transposase Affect the Transposition Activity of Bamboo Ppmar2 Mariner-Like Element"

_ijms, 2019, doi:10.3390/ijms20153692_

Round 1

Reviewer 1 Report

The authors characterize Mariner-like elements (MLEs) in bamboo and focus on Ppmar2 to determine the sequence specificity of the Terminal Inverted Repeats (TIR) for the DNA binding Domain of the encoded transposase and the effect on transposition.

Introduction:

The introduction is thorough and gives a good overview of the current state of the field.

-       Line 68 has a typo where the ‘a’ in ‘as’ should be capitalized.

-       The paragraph starting line 100 – 112 makes it sound as if the described work is part of the current manuscript. It should be rephrased to make it clear that this is prior work from the authors and indeed only the paragraph starting in line 113 – 121 describes the current work.

Results:

-       Line 130 refers to a T in the Box II sequence which is stated as GGCG just above. Please clarify.

-       Line 133-134: According to Table 1, 19 nucleotides were mutated including one G which is not stated in the text. Please correct either the text or Table 1 depending on what has been performed.

-       Line 148-149: Where is the data supporting the saturation of binding?

-       Line 155-157: ‘Affinities … significantly decreased’. It is not stated how this was quantified. Was there an actual quantification of signal detected across lane or in the non-bound fraction or is this simply based on visual inspection? In the absence of any statistical testing of quantifiable results, the use of the word significant should be avoided.

-       Line 160-161: Inconsistency in labeling between Phmar2-DBD and Ppmar2-DBD. Is this intentional? If so, please state what it means. If not, please refrain to using only one spelling. This is not an isolated incidence, but also occurs in Figure 2, Table 1, Figure 5 and Figure 6 (as examples, this list is not exhaustive).

-       Line 191-192: How is this being assessed? What is the read out? Please state here in the text.

-       Line 193-194: This statement is not correct. Similar results were only observed for TIR6 using Ppmar1 whereas TIR3 showed no difference to wildtype.

-       Line 203: ‘Statistically very significant’ – What statistical test was used and what was the p-value threshold to call something statistically very significant?

Discussion:

-       Line 216: Mutations to Box I (in TIR3) only showed a difference in the transposase assay

-       Line 220-230 – This paragraph describes previous study and is not relevant to the current discussion. It should be shortened or removed.

-       Line 231-234 – Why are the results not show? And where is it described how the sub-terminal TIRs are deleted? Please ensure the reproducibility of this study by including all necessary information.

-       Line 234-235: This sentence is more suitable for the Results section. Such clear interpretation of the results is currently lacking from the results part and would be beneficial even if repeated but slightly rephrased.

-       Line 244-246: This sentence is more suitable for the Results section (see comment above).

Figures and tables :

-       Figure 2 – For clarity, print the consensus sequence for Box 1 and Box 2 above

-       Figure 3 is lacking a size marker.

-       Figure 3 – please comment why there is no clear shift observed, but instead a decrease in signal. It is not stated in the Figure legend that the concentration of Ppmar2NA-TIRs is kept consistent across the titrations of the DBD.

-       Table 2 – The interaction  of TIR6 and TIR 7 was scored equally (“++ medium interaction”) for both 120uM and 150 uM DBD concentration, whereas 140uM was scored as high interaction. Given the gel picture, I fail to understand how 120uM and 150uM can be scored the same. Please clarify on the quantification being used here.

-       Figure 4 – Similar to comment on Figure 3. Please clarify, how the lanes with the least signal but no visible shift have been characterized as the highest interaction.

-       Figure 5 – If this is based on 6 replicates, why are there no error bars?

Methods:

-       Line 262: What was used for alignment?

-       Section 4.7: Please clarify how the DBDs were purified.

Author Response

Response to Reviewer 1 Comments

Point 1: Extensive editing of English language and style required.

Response 1: We have now edited the entire manuscript for better clarity and scientific style.

Point 2: The authors characterize Mariner-like elements (MLEs) in bamboo and focus on Ppmar2 to determine the sequence specificity of the Terminal Inverted Repeats (TIR) for the DNA binding Domain of the encoded transposase and the effect on transposition.

Response 2: We appreciate the constructive summary from the reviewer.

Introduction:

Point 3: The introduction is thorough and gives a good overview of the current state of the field.

Response 3: We thank the reviewer for positive comment.

Point 4:  - Line 68 has a typo where the ‘a’ in ‘as’ should be capitalized.

Response 4: We thank for pointing out this error in our manuscript. Line 68, where the ‘a’ in ‘as’ (As) has been capitalized. 

Point 5: The paragraph starting line 100 – 112 makes it sound as if the described work is part of the current manuscript. It should be rephrased to make it clear that this is prior work from the authors and indeed only the paragraph starting in line 113 – 121 describes the current work.

Response 5: We have completely revised the paragraph for improving clarity. Please see line 68.

Point 6: Results: - Line 130 refers to a T in the Box II sequence which is stated as GGCG just above. Please clarify.

Response 6: We thank the reviewer for pointing out this. The “T” has been deleted in the revised manuscript. Please see line 131.

Point 7: - Line 133-134: According to Table 1, 19 nucleotides were mutated including one G which is not stated in the text. Please correct either the text or Table 1 depending on what has been performed.

Response 7: We thank the reviewer for pointing out this. One “G” was also mutated. This is now added in the text. Please see line 135.

Point 8: - Line 148-149: Where is the data supporting the saturation of binding?

Response 8: We thank the reviewer. The data is not shown in the manuscript. The word “data is not shown” has been added in the text. Please see lines 149-153.   

Point 9: - Line 155-157: ‘Affinities … significantly decreased’. It is not stated how this was quantified. Was there an actual quantification of signal detected across lane or in the non-bound fraction or is this simply based on visual inspection? In the absence of any statistical testing of quantifiable results, the use of the word significant should be avoided.

Response 9: We thank the reviewer for valuable comments. Based on visual inspection, the affinities were observed. The word “significantly” has been removed from the revised manuscript. Please see line 159.

Point 10: -   Line 160-161: Inconsistency in labeling between Phmar2-DBD and Ppmar2-DBD. Is this intentional? If so, please state what it means. If not, please refrain to using only one spelling. This is not an isolated incidence, but also occurs in Figure 2, Table 1, Figure 5 and Figure 6 (as examples, this list is not exhaustive).

Response 10: We thank the reviewer for pointing out these errors. Ppmar2-DBD is correct terminology. Only Ppmar2-DBD has been used in the revised manuscript. Phmar2-DBD and Phamr2 have been removed from the text, Figure 2, Figure 5, Figure 6 and Table 1.

Point 11: - Line 191-192: How is this being assessed? What is the read out? Please state here in the text.

Response 11: We thank the reviewer for valuable comments. Transposition activity was assessed by a number of yeast colonies grown on medium CSM-ade-his-ura with 2% galactose. This is now added in the text, line 355.  

Point 12: - Line 193-194: This statement is not correct. Similar results were only observed for TIR6 using Ppmar1 whereas TIR3 showed no difference to wildtype.

Response 12: We thank the reviewer for the correct statement. Lines 193-194 have been revised. “Similar results were only observed for Ppmar2NA-TIR6 using Ppmar1 transposase whereas Ppmar2NA-TIR3 showed no difference to wildtype Ppmar2NA-TIR1 (Figure 5)”.

Point 13: - Line 203: ‘Statistically very significant’ – What statistical test was used and what was the p-value threshold to call something statistically very significant?

Response 13: The transposition activity was determined by counting the number of colonies on the plate and using SPSS software. The p-value was P<0.01. The p-value has been added in the revised manuscript. Please see line 214.

Discussion:

Point 14: -Line 216: Mutations to Box I (in TIR3) only showed a difference in the transposase assay

Response 14: We are very sorry, we could not understand the reviewer question. We feel that the statement is correct.

Point 15: - Line 220-230 – This paragraph describes previous study and is not relevant to the current discussion. It should be shortened or removed.

Response 15: Lines 220-230 have been revised and shortened in the revised manuscript.

Point 16: - Line 231-234 – Why are the results not show? And where is it described how the sub-terminal TIRs are deleted? Please ensure the reproducibility of this study by including all necessary information.

Response 16: We thank the reviewer. We feel that it is not necessary to add the deletion details of sub-terminal TIRs.

Point 17: - Line 234-235: This sentence is more suitable for the Results section. Such clear interpretation of the results is currently lacking from the results part and would be beneficial even if repeated but slightly rephrased.

Response 17: We appreciate the reviewer for thoughtful comments. “In the current project, we deleted the sub-terminal sequences of TIRs of Ppmar2NA to overcome the difficulty in the yeast transposition assay”; the slightly rephrased sentence has also been added in the results section. Please see lines 193-195

Point 18: - Line 244-246: This sentence is more suitable for the Results section (see comment above).

Response 18: We appreciate the reviewer for thoughtful comments. “and more frequent transposition was catalysed by either Ppmar1 transposase or Ppmar2 transposase”; the slightly rephrased sentence has also been added to the results section. Please see line 201.

Figures and tables:

Point 19: - Figure 2 – For clarity, print the consensus sequence for Box 1 and Box 2 above

Response 19: Figure 2 has been revised for better clarity.

Point 20: - Figure 3 is lacking a size marker.

Response 20: We thank the reviewer. We are very sorry. Unfortunately, we do not have that.

Point 21: - Figure 3 – please comment why there is no clear shift observed, but instead a decrease in signal. It is not stated in the Figure legend that the concentration of Ppmar2NA-TIRs is kept consistent across the titrations of the DBD.

Response 21: We appreciate the suggestion of the reviewer. We could not observe a clear shift due to high affinity.  

Point 22: - Table 2 – The interaction of TIR6 and TIR7 was scored equally (“++ medium interaction”) for both 120uM and 150 uM DBD concentration, whereas 140uM was scored as high interaction. Given the gel picture, I fail to understand how 120uM and 150uM can be scored the same. Please clarify on the quantification being used here.

Response 22: We appreciate the reviewer for thoughtful comments. We observed that the level of smear (affinity) was equal in TIR6 and TIR7 (“++ medium interaction”) for both 120uM and 150 uM DBD concentration. Further, we had only three groups; N no interaction: ++ medium interaction: *** high interaction.

Point 23: - Figure 4 – Similar to comment on Figure 3. Please clarify, how the lanes with the least signal but no visible shift have been characterized as the highest interaction.

Response 23: We appreciate the reviewer for kind comments. We could not observe the visible shift due to the high affinity of DBD with Ppmar2NA-TIRs.

Point 24: - Figure 5 – If this is based on 6 replicates, why are there no error bars?

Response 24: The reviewer comment is well taken. We have significantly revised the figure 5 and error bars have been added in the revised figure. We hope that the revised figure would be satisfactory the reviewer.

Methods:

Point 25: - Line 262: What was used for alignment?

Response 25: For alignment TIRs of multiple MLE transposons was used.

Point 26: - Section 4.7: Please clarify how the DBDs were purified.

Response 26: The transposase DBD short peptide was synthesized and purified by Shenggong Bioengineering Co., Ltd.

Reviewer 2 Report

The main problem of the manuscript is that in the introduction they suggest the possible use of transposons as generators of genetic variability in bamboo and, however, the experimental tests are not made in bamboo but in yeast. I do not think it is necessary for the authors to improve this part to accept the article, but they should make it clear in the discussion that the next step should be to introduce the system of modified transposons in bamboo.

Anyway, the results are interesting and they are done correctly, so I think they have to be published.

Author Response

Response to Reviewer 2 Comments

Point 1: English language and style are fine/minor spell check required.

Response 1: We have checked the entire manuscript for spelling.

Point 2: The main problem of the manuscript is that in the introduction they suggest the possible use of transposons as generators of genetic variability in bamboo and, however, the experimental tests are not made in bamboo but in yeast. I do not think it is necessary for the authors to improve this part to accept the article, but they should make it clear in the discussion that the next step should be to introduce the system of modified transposons in bamboo.

Response 2:  We appreciate the reviewer for the very positive, constructive summary for our manuscript. We have clearly discussed future studies in line 265-270; “These results, however, for the broader biological implications of Ppmar2NA-TIRs need to be addressed in future investigations. Furthermore, to validate the hyperactivity of mutated Ppmar2NA-TIRs, they need to be examined in model plants, such as foxtail millet or rice or Arabidopsis. This could help in developing actively modified transposons for tools of genetic manipulations and bamboo improvement” in the discussion section. Please see lines 266-270.

Point 3: Anyway, the results are interesting and they are done correctly, so I think they have to be published.

Response 3: We sincerely thank the reviewer for recommending our manuscript for publication in “International Journal of Molecular Science (IJMS)”. 

Reviewer 3 Report

In the manuscript "Affinities of Terminal Inverted Repeats to DNA binding domain of transposase affect the transposition activity of bamboo Ppmar2 mariner-like element", Ramakrishnan et al. analyzed the binding activity of the bamboo MLE transposases, Ppmar1 and Ppmar2, to seven mutated terminal inverted repeat (TIR) sequences. Two of the TIR mutants showed more string binding activity to the transposases in vitro, and one TIR indeed showed a higher transposition frequency in yeast.

The aim of this work is clear, and the authors successfully identified a TIR sequence that has a high transposition activity. This result may contribute to the development of a new gene tagging tool in bamboo. I have some minor points that the authors need to be addressed.

1) In Abstract and Introduction, the authors described about "sub-TIRs", but was not shown in Figure 1 or elsewhere. Please show the location of sub-TIRs in Figure 1 if Ppmar2 has sub-TIR sequence.

2) Page 2, line 68; "replaced by as" --> "replaced by As" ?

3) Page3, line 111-112; "The artificial-constructed transposase variant S171A was10-fold more active than the wild type transposase."

A reference should be provided for these sentences. Or, unpublished data?

4) Page 5, lines 148-149; "The binding of Ppmar2NA-TIRs was saturated when Ppmar2-DBD concentration was more than 200 μM."

"Data not shown" should be added.

4) Page 5, lines 160-161; "These results indicated that the A and T affected the TIRs binding capacity ..."

Readers may confuse which A and T in Table 1 the authors refer to. I recommend that authors provide site numbers in Figure 2.

5) Page 5, line 175; "Ppmar2NA-TIRs had a very weak affinity to Ppmar1-DBD."

From this result, it is expected that the transposition frequency by Ppmar2 is higher than that by Ppmar1 in the yeast transposition assay. However, both the transposition frequency are normalized to 1.0 in Figure 5 (left). Please show the difference of the transposition frequency of the wild-type Ppmar2NA-TIR1 between Ppmar1 and Ppmar2 transposases.

6) Throughout the manuscript, Ppmar2 is sometimes written as "Phmar2". Please check.

7) Table 1;

The authors should highlight Box1 and Box2 in the sequences because it was not easy for me to find the boxes. I recommend that this table is shown as a part of Figure 2.

8) Figure 1;

The TIR (blue triangles), TSD (red bar), and Exons (yellow box) are separately represented. Please show them together. The two "TIRs" in the left and right sides of Ppmar2 Transposon are unnecessary.

9) Figure 2;

The eight sequences shown are not consensus sequences. However, I consider that consensus sequences should be compared in such a case because a sequence from only one locus may not represent the original sequence of each of the MLE families due to substitutions after integration.

10) Figure 5;

They described that "The symbol ** indicates statistically very significant difference with the wild type Ppmar2NA-TIR1." How significant (p-value)? Which statistical test did the authors use? In addition, please show error bars for the graph.

Author Response

Response to Reviewer 3 Comments                                                 

Point 1: Moderate English changes require.

Response 1: We have now revised the entire manuscript for better clarity and scientific style.

Point 2: In the manuscript "Affinities of Terminal Inverted Repeats to DNA binding domain of transposase affect the transposition activity of bamboo Ppmar2 mariner-like element", Ramakrishnan et al. analyzed the binding activity of the bamboo MLE transposases, Ppmar1 and Ppmar2, to seven mutated terminal inverted repeat (TIR) sequences. Two of the TIR mutants showed more string binding activity to the transposases in vitro, and one TIR indeed showed a higher transposition frequency in yeast.

Response 2: We thank fo the constructive comments and suggestions from the reviewer.

Point 3: The aim of this work is clear, and the authors successfully identified a TIR sequence that has a high transposition activity. This result may contribute to the development of a new gene tagging tool in bamboo. I have some minor points that the authors need to be addressed.

Response 3: We thank the reviewer for the appreciation of our work.

Point 4: 1) In Abstract and Introduction, the authors described about "sub-TIRs", but was not shown in Figure 1 or elsewhere. Please show the location of sub-TIRs in Figure 1 if Ppmar2 has sub-TIR sequence.

Response 4: We appreciate the comments of the reviewer. In the current study, we studied only the affinities of Terminal Inverted Repeats (TIRs) to DNA binding domain (DBD) and their influences of the transposition activity of Ppmar2. Sub-terminal sequences of TIRs of Ppmar2NA affected the transposition of Ppmar2NA in the yeast. Thereby, sub-terminal sequences of TIRs of Ppmar2NA were deleted to meet the difficulty in the yeast transposition assay. We have removed the word “sub-TIRs” from the abstract. Please see lines 26-27.

Point 5: 2) Page 2, line 68; "replaced by as" --> "replaced by As"?

Response 5: We thank the reviewer for pointing out this error. Line 68, where the ‘as’ has been replaced by ‘As’. 

Point 6: 3) Page3, line 111-112; "The artificial-constructed transposase variant S171A was10-fold more active than the wild type transposase." A reference should be provided for these sentences. Or, unpublished data?

Response 6: The reviewer suggestion is well taken. We have revised the sentence. The following appropriate reference has also been added in the revised manuscript.  Please see line 112.

[15]. Zhou, M.B.; Hu, H.; Miskey, C.; Lazarow, K.; Ivics, Z.; Kunze, R.; Yang, G.; Izsvak, Z.; Tang, D.Q. Transposition of the bamboo Mariner-like element Ppmar1 in yeast. Mol Phylogenet Evol 2017, 109, 367-374, doi: 10.1016/j.ympev.2017.02.005.

Point 7: 4) Page 5, lines 148-149; "The binding of Ppmar2NA-TIRs was saturated when Ppmar2-DBD concentration was more than 200 μM." Data not shown" should be added

Response 7: We thank the reviewer. We feel that it is not necessary

Point 8: 4) Page 5, lines 160-161; "These results indicated that the A and T affected the TIRs binding capacity ..." Readers may confuse which A and T in Table 1 the authors refer to. I recommend that authors provide site numbers in Figure 2

Response 8: The reviewer suggestion is well taken. We have revised the sentences, the Table 1 and figure 2. We hope that the revised Table 1 and figure 2 make it more clear about the A and T.

Point 9: 5) Page 5, line 175; "Ppmar2NA-TIRs had a very weak affinity to Ppmar1-DBD." From this result, it is expected that the transposition frequency by Ppmar2 is higher than that by Ppmar1 in the yeast transposition assay. However, both the transposition frequency are normalized to 1.0 in Figure 5 (left). Please show the difference of the transposition frequency of the wild-type Ppmar2NA-TIR1 between Ppmar1 and Ppmar2 transposases.

Response 9: We appreciate the comments of the reviewer. We could not observe the difference in the transposition frequency for the wild-type Ppmar2NA-TIR1 between Ppmar1 and Ppmar2 transposases.

Point 10: 6) Throughout the manuscript, Ppmar2 is sometimes written as "Phmar2". Please check.

Response 10: We thank the reviewer. We thoroughly checked throughout the manuscript. Ppmar2 is the correct one, and Phmar2 has been removed.

Point 11: 7) Table 1; The authors should highlight Box1 and Box2 in the sequences because it was not easy for me to find the boxes. I recommend that this table is shown as a part of Figure 2.

Response 11: We appreciate the reviewer. Box1 and Box2 have been highlighted in Table 1.

Point 12: 8) Figure 1; The TIR (blue triangles), TSD (red bar), and Exons (yellow box) are separately represented. Please show them together. The two "TIRs" in the left and right sides of Ppmar2 Transposon are unnecessary.

Response 12: We have revised Figure 1 as per the reviewer suggestions. We hope that the revised figure is satisfactory.

Point 13: 9) Figure 2; The eight sequences shown are not consensus sequences. However, I consider that consensus sequences should be compared in such a case because a sequence from only one locus may not represent the original sequence of each of the MLE families due to substitutions after integration.

Response 13: We appreciate the comments of the reviewer. We take this suggestion into account in our ongoing experiment.

Point 14: 10) Figure 5; They described that "The symbol ** indicates statistically very significant difference with the wild type Ppmar2NA-TIR1." How significant (p-value)? Which statistical test did the authors use? In addition, please show error bars for the graph.

Response 14: We have revised Figure 5 as per the reviewer suggestions. We have shown the error bars in the revised figure. The p-value was P<0.01. The SPSS software was used for analysis. Please see line 214.

Reviewer 4 Report

ijms-529857-peer-review-v1

Full Title: Affinities of Terminal Inverted Repeats to DNA binding domain of transposase affect the  transposition activity of bamboo Ppmar2 mariner-like element

General comments to the manuscript:

The present manuscript deals with a study that aims to investigate the affinities of Terminal Inverted Repeats to DNA binding domain and their influences on the transposition activity of a specific DNA transposon (Ppmar2). To reach their goals, authors follow a site-directed mutagenesis approach, associated with gel mobility shift assays.

The manuscript is very well written, it comprises a good introduction, the results are very interesting and would merit of publication in the International Journal of Molecular Sciences journal; however, there are some few points that need to be improved before final acceptance. I would recommend to improve the manuscript making the changes suggested below.

I agree with minor revision.

Line 68: rephrase this sentence.

Line 82: rephrase sentence.

Line 100: change “in 38 genera” by “from 38 genera”

Line 109: “that” is repeated twice

Line 128: please confirm nucleotide sequence. I didn’t found the sequences here described as so conserved across all sequences. Next sentence related with nucleotide variability/conservation is for me also confusing, I would request authors to rewrite it.

Line 156: remove “detected”

Line 161: please confirm the results pointed out in this last sentence. Complete the sentence - affinity to what?

Line 161: remove “nucleotide”

Line 174: remove “was”

Line 262: remove “other” and add “sequences” after TIR (…were aligned with TIR sequences of…).

Line 266: rephrase sentence ”Drosophila mauritiana [51] and two conserved domains” like Drosophila mauritiana [51]. Two conserved domains”

Line 288: “were sequenced” how? In sense/antisense, with specific primers

Author Response

Response to Reviewer 4 Comments

Point 1: English language and style are fine/minor spell check required.

Response 1: We have now thoroughly checked the entire manuscript for spelling.

General comments to the manuscript:

Point 2: The present manuscript deals with a study that aims to investigate the affinities of Terminal Inverted Repeats to DNA binding domain and their influences on the transposition activity of a specific DNA transposon (Ppmar2). To reach their goals, authors follow a site-directed mutagenesis approach, associated with gel mobility shift assays.

The manuscript is very well written, it comprises a good introduction, the results are very interesting and would merit of publication in the International Journal of Molecular Sciences journal; however, there are some few points that need to be improved before final acceptance. I would recommend to improve the manuscript making the changes suggested below.

Response 2: We appreciate the reviewer for the very positive, constructive conclusion for our manuscript.

Point 3: I agree with minor revision.

Response 3: We sincerely thank the reviewer for endorsing our manuscript for publication.  

Point 4: Line 68: rephrase this sentence.

Response 4: This sentence has been rephrased in the revised manuscript.

Point 5: Line 82: rephrase sentence.

Response 5: This sentence has been rephrased into a past perfect sentence in the revised manuscript.  

Point 6: Line 100: change “in 38 genera” by “from 38 genera”

Response 6: “in 38 genera” has been changed into “from 38 genera”

Point 7: Line 109: “that” is repeated twice

Response 7: We thank the reviewer for pointing out this. “The repeated “that” has been deleted.  

Point 8: Line 128: please confirm nucleotide sequence. I didn’t found the sequences here described as so conserved across all sequences. Next sentence related with nucleotide variability/conservation is for me also confusing, I would request authors to rewrite it.

Response 8: We thank the reviewer for his interest. We have rewritten the sentences. 

Point 9: Line 156: remove “detected”

Response 9: We thank the reviewer for pointing out this. “The “detected” has been deleted.  

Point 10: Line 161: please confirm the results pointed out in this last sentence. Complete the sentence - affinity to what?

Response 10: We have rewritten the sentences. “The two bp mutations in Ppmar2NA-TIR6 and Ppmar2NA-TIR7 had a significant effect on the affinity to Ppmar2-DBD”. Please see lines 163-165.

Point 11: Line 161: remove “nucleotide”

Response 11: The “nucleotide” has been removed.  

Point 12: Line 174: remove “was”

Response 12: The “was” has been removed.  

Point 13: Line 262: remove “other” and add “sequences” after TIR (…were aligned with TIR sequences of…).

Response 13: The “other” has been removed, and “sequences” have been added after TIR.

Point 14: Line 266: rephrase sentence ”Drosophila mauritiana [51] and two conserved domains” like Drosophila mauritiana [51]. Two conserved domains”

Response 14: We have rephrased the sentences as per the reviewer suggestions.

Point 15: Line 288: “were sequenced” how? In sense/antisense, with specific primers

Response 15: To confirm the presence of the targeted mutation, all plasmids were sequenced with specific primers. This is now added in the text.

Round 2

Reviewer 1 Report

I thank the authors for addressing many of the points that I have raised during their revision of the manuscript.

However, there are still a few points where I am not satisfied with their response.

Point 8 regarding the missing data that demonstrates the saturation of binding. I don't understand why the authors are refusing to add this piece of data as a supplemental figure since it was also raised by another reviewer. This is a classic control experiment and should be displayed to help the authors interpret the blots. Especially in the absence of a clear gel shift, I think it would be helpful for readers to see what a full saturation of binding looks like in these experiments.

Point 16: I would have hoped that in the spirit of reproducibility the authors are willing to provide all necessary experimental details to allow readers to reconstruct and reproduce the experiments undertaken.

Point 22: I still find this characterisation troubling as these groups are based on visual inspection and are not quantified by signal intensity or the like. The separation in only three groups seems not sufficient to me as to my eye, 120uM and 150uM do not look the same for TIR6 and TIR7. In fact, to me, 150uM looks much more similar to 140uM and thus I would have expected the 150uM condition to be classified as a high interaction (same as 140uM) instead of a medium interaction. As it stands, I fail to understand how the authors perform the classification thus making it look arbitrary to me. 

Author Response

Point 1: English language and style are fine/minor spell check required.

Response 1: We have now edited the entire manuscript for better clarity and scientific style.

Point 2: I thank the authors for addressing many of the points that I have raised during their revision of the manuscript. However, there are still a few points where I am not satisfied with their response.

Response 2: We are very grateful for the reviews and appreciate the constructive summary from the reviewer. The comments are encouraging us.

Point 3: Point 8 regarding the missing data that demonstrates the saturation of binding. I don't understand why the authors are refusing to add this piece of data as a supplemental figure since it was also raised by another reviewer. This is a classic control experiment and should be displayed to help the authors interpret the blots. Especially in the absence of a clear gel shift, I think it would be helpful for readers to see what a full saturation of binding looks like in these experiments.

Response 3:

We have now quantified the electrophoretic patterns of the binding complexes in case of both Ppmar1 and Ppmar2 transposases. Quantification was done by image intensity analysis from the gel patterns. We have compared the binding intensity of the transposase domains to the TIRs based on the concentration of the substrates used for comparing the efficiency of mutant TIRs. Further, we have added an additional section on the affinity shift based on the type of nucleotide substitutions in the mutant TIRs. The data is now presented as graphs and tables and also discussed in detail. Although there was a initial setback in presenting this data in the earlier versions of the manuscript we have now resolved this. We are highly thankful for the reviewer for being insistent on this piece of information, which we now feel has added great value to our paper. 

Point 4: Point 16: I would have hoped that in the spirit of reproducibility the authors are willing to provide all necessary experimental details to allow readers to reconstruct and reproduce the experiments undertaken.

Response 4:

We have now elaborated all the experimental procedures and the means through which different analyses were done and presented in the paper. We have also added additional information to help the readers to better understand the experimental details as desired by the reviewer.

Point 5: Point 22: I still find this characterisation troubling as these groups are based on visual inspection and are not quantified by signal intensity or the like. The separation in only three groups seems not sufficient to me as to my eye, 120uM and 150uM do not look the same for TIR6 and TIR7. In fact, to me, 150uM looks much more similar to 140uM and thus I would have expected the 150uM condition to be classified as a high interaction (same as 140uM) instead of a medium interaction. As it stands, I fail to understand how the authors perform the classification thus making it look arbitrary to me.

Response 5: We have now addressed this issue completely as mentioned in the earlier response #3. We have now resolved the patterns between the concentrations and also between the transposases, including discussions on affinity shift based on the nucleotide substitution pattern.